# Cortisol on Circadian Rhythm and Its Effect on Cardiovascular System

**DOI:** 10.3390/ijerph18020676

**Published:** 2021-01-14

**Authors:** Nor Amira Syahira Mohd Azmi, Norsham Juliana, Sahar Azmani, Nadia Mohd Effendy, Izuddin Fahmy Abu, Nur Islami Mohd Fahmi Teng, Srijit Das

**Affiliations:** 1Faculty of Medicine and Health Sciences, Universiti Sains Islam Malaysia, Nilai 71800, Negeri Sembilan, Malaysia; amirasyahira188@gmail.com (N.A.S.M.A.); drazmanisahar@usim.edu.my (S.A.); nadia@usim.edu.my (N.M.E.); 2Institute of Medical Science Technology, Universiti Kuala Lumpur, Kajang 43000, Selangor, Malaysia; izuddin@unikl.edu.my; 3Faculty of Health Sciences, Universiti Teknologi MARA, Bandar Puncak Alam 42300, Selangor, Malaysia; nurislami@uitm.edu.my; 4Department of Anatomy, Universiti Kebangsaan Malaysia Medical Centre, Cheras 56000, Kuala Lumpur, Malaysia; srijit@ukm.edu.my

**Keywords:** cortisol, biological clock, circadian rhythm, heart, cardiovascular

## Abstract

The synthesis and secretion of cortisol are controlled by the hypothalamic–pituitary–adrenal axis. Cortisol exhibits a proper 24-h circadian rhythm that affects the brain, the autonomic nervous system, the heart, and the vasculature that prepares the cardiovascular system for optimal function during these anticipated behavioral cycles. A literature search was conducted using databases such as Google Scholar, PubMed, and Scopus. Relevant search terms included “circadian rhythm and cardiovascular”, “cortisol”, “cortisol and acute coronary syndrome”, “cortisol and arrhythmias”, “cortisol and sudden cardiac death”, “cortisol and stroke”, and “cardioprotective agents”. A total of 120 articles were obtained on the basis of the above search. Lower levels of cortisol were seen at the beginning of sleep, while there was a rise towards the end of sleep, with the highest level reached at the moment the individual wakes up. In the present review, we discuss the role of 11β-hydroxysteroid dehydrogenase (11β-HSD1), which is a novel molecular target of interest for treating metabolic syndrome and type-2 diabetes mellitus. 11β-HSD1 is the major determinant of cortisol excess, and its inhibition alleviates metabolic abnormalities. The present review highlights the role of cortisol, which controls the circadian rhythm, and describes its effect on the cardiovascular system. The review provides a platform for future potential cardioprotective therapeutic agents.

## 1. Introduction

Circadian rhythm is known as a periodic pattern that takes about 24-h, where the light–dark cycles synchronize biological functions with the environment [1]. The circadian system coordinates physiology and behaviors towards the environment in such a way that the body acts like a finely harmonized clock. The suprachiasmatic nucleus (SCN) of the hypothalamus functions as the master of the clock, synchronizing 24-h rhythms in the body’s physiological behavior, including other brain regions and peripheral tissues. Identical clock oscillators to the SCN clock have been established in peripheral tissues such as gastrointestinal tract, liver, muscle, adipose tissue, and cardiovascular tissue. The clock induces sleep and other related anabolic functions at night when synchronized according to the environment, such as immune function and hormone release, and wakefulness and its associated catabolic functions throughout the day, i.e., food intake and metabolism and physical activity [2]. One of the most potent hormones in human physiology is cortisol, in which nearly all of the cells of the body are potential cortisol targets. It provides one of the ways of transmitting the circadian message from the SCN to the peripheral tissues. The cortisol peak after waking can play a specific role in synchronizing the body across a range of nongenomic acts to both the sleep–wake and light–dark cycles [3]. In addition, it is understood that circadian rhythms transcribe the message of the time of day to the immune system, especially that of cortisol. For good physical and mental well-being, circadian coordination is essential, and disturbance of circadian activity is correlated with various downstream negative physiological, psychological, and clinical implications [4]. The present review discusses the role of cortisol in circadian rhythm regulation with special emphasis on the cardiovascular system.

Cortisol has a particular circadian rhythm that is affected by sleep. Based on normal physiology, lower levels of cortisol are present at the initial part of sleep, whereas there is an increase at the end of the sleep period, which reaches its peak just minutes before the individual wakes up. This rhythm is formed in close association with the sleep–wakefulness cycle [5]. Thus, it supports the fact that cortisol contributes to a significant role in the initiation of wakefulness [6]. In other words, cortisol is the key player in the circadian system that affects nearly every tissue and organ of the body, controlling certain biological cyclical functions, including cardiovascular functions [7]. Distinct variation in the levels of cortisol during the day and night may help to explain the function of cortisol in health and disease [8]. Cortisol status appears to be an objective biological marker of the stress response that can be related to the majority of cardiovascular diseases [9]. The timing of cardiovascular events like acute myocardial infarction and ventricular fibrillation can influence the severity of disease and survival, with poor prognoses being associated with early morning occurrences [10]. In the general population, the levels of cortisol measured in blood, urine, or hair are positively associated with elevated risk factors for cardiovascular disease [11], death from cardiovascular disease, and all-cause mortality [12].

## 2. Methodology

The literature search for relevant articles of this narrative review was conducted in March 2020 using the Google Scholar, PubMed, and Scopus databases. The studies were identified based on the details in the title and abstract. Relevant search terms included “circadian rhythm and cardiovascular”, “cortisol”, “cortisol and acute coronary syndrome”, “cortisol and arrhythmias”, “cortisol and sudden cardiac death”, “cortisol and stroke”, and “cardioprotective agents”. We managed to retrieve 120 articles on the basis of the above search. All the studies were limited to journals in the English language, and we considered all published articles from 2000 to 2020.

## 3. Discussion

This section outlines the circadian rhythm and its cardiovascular function, cortisol and its physiology, and the effects of cortisol on the cardiovascular system and drug delivery.

### 3.1. Circadian Rhythm and Cardiovascular Functions

#### 3.1.1. The Biological Clock that Controls the Cardiovascular System (CVS)

Almost all cardiovascular parameters fluctuate throughout the day, with unique patterns between day and night. It is of great significance to assess blood pressure (BP), heart rate (HR), blood coagulation markers, vascular endothelial function, and circulating catecholamines at the right time of the day [13]. Highlighting the unique characteristics of these parameters, the advent of 24-h fluctuation standard ranges based on earlier studies has allowed continuous and dynamic circadian physiological signals to be monitored nowadays. Circadian blood pressure and heart rate are direct reflections of the relative integrity of the autonomic nervous system [14,15]. Intact autonomic function demonstrates that there is a correlation between heart rate and peripheral blood pressure; an inverse relationship between heart rate and central blood pressure exists [16]. The circadian rhythm of platelet count warrants serious attention as it is one of the most important parameters in the coagulation system. Barceló et al. (2011) observed that platelet activity is very high around 06:00 a.m., upon rising from sleep, and it fluctuates throughout the day, with the peak of its activity approximately at 11:00 p.m. [17]. Any alterations of these important parameters further affect cardiovascular circadian function.

CVS 24-h regulation is complicated as it is controlled by the interaction between environmental and intrinsic factors. The biological clock that exists in cardiomyocytes, vascular smooth muscle, and endothelial cells highly influences CVS circadian rhythm by manipulating the autonomic nervous system. The clock ensures rapid and appropriate responses between sympathetic and parasympathetic signals throughout the day [18,19]. Time-of-day-dependent oscillations in the physiology and pathology of the cardiovascular system are attributed to fluctuations in numerous neurohumoral factors. Furthermore, these include sympathetic–autonomic–adrenergic stimulation, endocrine factors such as cortisol, nutrients, pro- and antithrombolytic factors, and vascular resistance. Nevertheless, evidence indicates that besides these extracellular factors, distinct cardiac processes are directly regulated in a time-of-day-dependent manner by intracellular mechanisms [20,21]. Figure 1 illustrates the biological clock in heart, digestive tract, liver, muscle, and immune cells.

#### 3.1.2. Circadian Changes in Pathophysiological Mechanisms of Cardiovascular Events

Generally, human sleep is classified into three stages, i.e., wake, rapid eye movement (REM) sleep, and nonrapid eye movement (NREM) sleep [22]. NREM sleep comprises four substages (Stages 1–4). The sleep stages of Stage 1 and Stage 2 are termed light sleep phases, while Stage 3 and Stage 4 are referred to as deep sleep phases [23]. The pattern of changes in blood pressure and heart rate fluctuations are related with the sleep–wake cycle, in which the specific stage of sleep is closely linked. Sleep is strictly correlated with the function of the autonomic nervous system [24]. Blood pressure and heart rate steadily decrease during the NREM sleep period from Stages 1 to 3 because of a gradual decline in sympathetic activity and the prevalence of parasympathetic tone. Meanwhile, at the time of REM sleep, parasympathetic tone still dominates, but periodic bursts of sympathetic activity result in distinct variations in blood pressure and heart rate [25]. Figure 2 shows the circadian rhythm of blood pressure and heart rate. Since REM sleep is more common in the early hours of the morning, these hemodynamic changes are causally associated with the highest proportions of cardiovascular and cerebrovascular events, including ischemic and hemorrhagic stroke, myocardial infarction, and sudden cardiac death [26].

Adverse cardiac events have often been reported to be found in the morning; therefore, the events exhibit a clear circadian pattern. Incidences such as acute myocardial infarction, sudden cardiac death, atrioventricular block, and ventricular fibrillation were approximately 40% higher between 06:00 a.m. and 12:00 p.m. in comparison to the rest of the day [28]. In addition, the peripheral clock system within each cardiovascular system tends to play crucial roles during the development of cardiovascular events. The progression of the disease may be aggravated when the phase of the central clock is not harmonized with that of the peripheral clock [19]. Sleep disorders, shift work, and jetlag are examples of circadian disruption [29]. In these populations, the cardiovascular system functions with a 24-h rhythm, like blood pressure and heart rate, and rhythms in core temperature and hormone secretion and activity are disturbed [30]. Furthermore, the phenomenon of circadian rhythm misalignment when the individual is involved in shift duties can consequently lead to an increased risk of metabolic syndrome [2]. Physiological reduction in cardiovascular rhythms is a bad prognostic indicator that is related to dysfunction of organs, delirium, and a high risk of death [31]. Apart from that, decreased heart rate variability that is associated with adiposity may develop at an early age, indicating that the recent high prevalence of obesity has serious implications for early cardiovascular events in our population [32].

Moreover, circadian rhythms are linked with the incidence, pathophysiology, and outcome of cardiovascular disease. A good example is myocardial infarction, which has a 24-h pattern [33,34]. Such changes are typically due to circadian variability of the autonomic nervous system [35]. Sympathetic activity, shear stress, and cardiovascular risk factors, including blood pressure, are certainly elevated in the morning, possibly triggering the onset of disease [36]. Moreover, the aggregation and coagulation of platelets, ventricular repolarization disorders, and many other related cardiovascular risk factors controlled by peripheral circadian clocks logically lead to a peak in the occurrence of cardiovascular events at different times of the day [37]. When myocardial infarction happens early in the morning, the resulting damage and cardiac dysfunction are worse compared to the same infarction occurring in the afternoon, even though not all studies show similar findings [38]. Thus, both time of onset and clock desynchronization after an infarction have significant effects on the outcome of cardiovascular events [39].

### 3.2. Cortisol and Its Physiology

#### 3.2.1. Synthesis and Metabolism of Cortisol

A significant component of the human neuroendocrine system is the hypothalamic–pituitary–adrenal gland (the HPA axis), which is of paramount importance to the survival of mammals, including humans [40]. The physiological function of the HPA axis is important to preserve homeostasis, adapt to environmental surroundings, and regulate human behavior, emotion, and cognitive functions. The main end product of the HPA axis is cortisol, a glucocorticoid hormone that is highly associated with circadian rhythms [41]. Cortisol is considered the major glucocorticoid in both humans and nonhuman primates, and it is produced in response to a variety of stressors [42].

Cortisol plays important roles in various physiological functions, such as metabolism, electrolyte balance, development, and cognition, and is implicated in multiple organ systems, namely, neuroendocrine, immune, reproductive, cardiovascular, and nervous systems [43,44]. Cortisol and other glucocorticoid hormones are mainly derived from cholesterol in the mitochondria of zona fasciculata in the adrenal cortex [45]. Metabolically, cholesterol is first converted to pregnenolone by removing the side chain at C20, a process catalyzed by desmolase (cytochrome P450scc). Pregnenolone is subsequently converted to progesterone by 3β-hydroxysteroid dehydrogenase. Enzyme 17-hydroxylase (CYP17) catalyzes the conversion of progesterone to 17-hydroxyprogesterone, which is then converted to 11-deoxycortisol by the 21α-hydroxylase activity of CYP21. The final step involves the 11β-hydroxylation of 11-deoxycortisol to the end product, cortisol, by 11-hydroxylase (CYP11B1 or CYP11B2) [43,46].

#### 3.2.2. Regulation and Implication of Cortisol Circadian Rhythm

The synthesis and secretion of cortisol are controlled by the HPA axis. Under normal conditions, without any stressors, cortisol is secreted by the adrenal glands into the bloodstream in a circadian and ultradian rhythm. In humans, the peak level secretion occurs in the morning (07:00–08:00 a.m.), which is considered the active phase, while its lowest secretion is around 02:00–04:00 a.m. at night [44,47]. Figure 2 demonstrates the circadian rhythm of cortisol. This periodic secretion is controlled by the circadian clock, a natural body time-keeper that involves a central pacemaker in the hypothalamus, the suprachiasmatic nucleus, and the adrenal gland clock itself, which controls the sensitivity of the gland to the adrenocorticotrophic hormone (ACTH). The circadian release of cortisol plays a part in the synchronization of the cell-autonomous clocks in the body and interacts with them to time physiological dynamics in their target tissues around the day [48].

HPA axis activity is also induced when there is a physiological response (such as a response by the immune system) as well as psychological and emotional stress. The activation of the HPA axis releases the corticotropin-releasing hormone (CRH) and arginine vasopressin (AVP) from the hypothalamic paraventricular nucleus, which then bind to their receptors (CRH-R1 and V1B, respectively) in the anterior pituitary. This binding triggers the secretion of the adrenocorticotrophic hormone (ACTH) in the bloodstream, which subsequently induces the adrenal gland to synthesize and release cortisol hormones [49]. The regulation of cortisol synthesis is controlled by a negative feedback mechanism mediated by mineralocorticoid and glucocorticoid receptors in the hypothalamus and anterior pituitary, whereby elevated levels of serum cortisol repress CRH, AVP, and ACTH release [43]. In contrast, extremely low levels of cortisol triggers enhanced production of CRH and ACTH [47].

Furthermore, internal biological day and night are demarcated by endogenous melatonin rhythm that is regulated by the SCN clock. In humans, during the biological night, there is a high level of melatonin secretion; low levels of melatonin are present during the biological day [50,51]. Melatonin triggers sleepiness in a dark environment of a typical physiological setting, and cortisol secretions sustain daytime consciousness throughout the morning [52]. Diurnal rhythm, consciousness, and the sleep–wake cycle, along with neural pressure signals, are factors affecting cortisol secretion. The secretion of normal cortisol follows a negative slope. During Stage 1 of sleep, cortisol secretion is lower, and it rises progressively during Stage 2 of sleep [53]. Based on normal physiology, when we wake up in the morning, the highest concentration of cortisol is achieved [54]. Normal levels of serum cortisol (approximately 250–850 nmol/L between 8:00 and 10:00 a.m.) retain daytime consciousness and then decline gradually throughout the day. At night, the concentration of cortisol is approximately half the concentration during the day, with a normal range of 110–390 nmol/L and the lowest levels at around 04:00 a.m. [55,56]. The range of cortisol concentration variation is stable around the clock. However, when a person is under stress or pressure, a rise in the secretion of cortisol can be observed [53,55].

Cortisol rhythmicity is crucial for the normal function of the HPA axis; it is not only influenced by external factors such as sleep and season but also by stress factors such as mental, psychological, or physical status [41]. Stress factors, especially during the “fight or flight” response, will impact plasma cortisol levels [57]. Irregular cortisol as a result of HPA axis impairments may, in turn, inordinate circadian rhythm and pose health risks on human emotion and behavior and physical and cognitive capabilities [41]. Reduced levels of cortisol as a result of adrenal insufficiency and ACTH deficiency, such as in congenital adrenal hyperplasia conditions, may consequently result in Addison’s disease [46] and can propagate other neurological disorders such as attention deficit hyperactivity disorder (ADHD) [58,59,60,61], dyslexia [62], and autism [63]. In contrast, elevated cortisol levels are toxic to the hippocampus, interfere with adaptive processes [64], and are associated with depression [42,65]. Excess cortisol as a result of circadian rhythm impairments, coupled with a loss of the HPA axis negative-feedback mechanism, may lead to the manifestation of Cushing’s syndrome [46]. Hypercortisolism has also been shown to trigger insulin resistance in tissues and organs, namely, liver, adipose, and skeletal muscle [66]. In fetal development, although cortisol is pivotal for lung maturation, development of the central nervous system, and stimulation of important enzymes, studies have reported that increased levels of prenatal cortisol may retard fetal growth [57].

### 3.3. Effect of Cortisol on the Cardiovascular System

#### 3.3.1. Acute Coronary Syndromes

Acute coronary syndromes (ACSs) are characterized as ST-segment elevation of myocardial infarction (STEMI), non-ST-segment elevation of myocardial infarction (NSTEMI), and unstable angina (UA), which lead to morbidity and mortality in industrialized nations [67]. In smaller cross-sectional studies, there have been contradictory findings of the relationship between cortisol and adverse cardiovascular outcomes in AMI patients, despite the majority showing a poorer outcome with increased levels of cortisol [68]. Specifically, most of the smaller cross-sectional studies have reported increased morbidity and mortality risks with elevated levels of cortisol in AMI patients [69,70], whereas some found no correlation [71] and others found an inverse association [72].

Moreover, previous studies have also reported the association of levels of cortisol with an increase in myocardial infarct size [73], ventricular remodeling post-acute myocardial infarction (AMI) [74], and high mortality in patients with chronic heart failure [75]. Jutla et al. (2014) found the prognostic importance of cortisol in post-AMI patients at a single center with respect to major adverse cardiovascular events (MACEs) and a combination of all-cause mortality, as well as rehospitalization for heart failure immediately following myocardial necrosis [68]. Meanwhile, a case–control study by Reynolds et al. (2010) found a reverse association in which lower serum cortisol levels in post-AMI patients were associated with increased 30-day mortality [72].

#### 3.3.2. Atrial and Ventricular Arrhythmias

Cardiac arrhythmias are described as the conduction of irregular electrical impulses via the myocardium, leading to altered heartbeat, altered muscle contraction, and disrupted rhythm [76]. With respect to the pathophysiology of arrhythmias and sudden cardiac death, an acute precipitating stimulus in the neurons of the brain and chronic electrical instability of the cardiomyocyte in the heart may be correlated with an increase in circulating cortisol [77].

Atrial fibrillation (AF) is the commonest type of arrhythmia in adults and a significant cause of morbidity and death [78]. Specifically, AF is an atrial tachyarrhythmia that is categorized by irregular atrial activation and impaired atrial contractile function [78]. Cortisol is a steroid hormone that enhances in response to stress and plays an important part in the synthesis of catecholamine. Catecholamine has arrhythmic potential by influencing the cardiac conduction system [79]. Glucocorticoids are also involved in the synthesis of Na-ATP, K-ATP, and catecholamine. Furthermore, in this regard, it is responsible for a partial positive inotropic impact on the heart [80]. However, in a case–control study by Akseli et al. (2013), which aimed to determine whether serum cortisol levels were different in AF patients in comparison to a control group, it was reported that the abnormality in the levels of serum cortisol with respect to diurnal rhythm did not correlate with the etiology and permanence of AF [81].

Ventricular tachycardia and ventricular fibrillation occur as a result of dynamic and complex interactions between the arrhythmogenic substrate (i.e., myocardial scar boundary zone) and myocardial electrophysiological properties (i.e., electrical stimulation and propagation) [82]. Previous studies have reported that repeated bouts of high cortisol in view of psychological stress or neural mechanisms can predispose to low-grade inflammation, causing poor processes of repair, with low macrophages and excess neutrophils and lymphocytes. Thus, this subsequently results in myocardial dysfunction-induced arrhythmias [83,84]. Moreover, any inflammatory focus in the myocardium due either to an acute increase in cortisol or myocardial ischemia may inhibit the myocardial cell coordination, leading to ventricular fibrillation [85].

#### 3.3.3. Sudden Cardiac Death

Sudden cardiac death remains a significant health concern that is responsible for a large proportion of all cardiac deaths [86]. A high incidence of sudden cardiac death may result from coronary heart disease, left ventricular hypertrophy, cardiac fibrosis, and heart failure [87]. In addition, arrhythmias also may result in sudden cardiac death due to focal or general inflammation induced by high concentrations of circulating cortisol [77]. According to the World Health Organization, sudden cardiac death is defined as sudden and unexpected death observed within an hour of symptom onset [88]. The underlying myocardial electrophysiological alteration in most patients with sudden cardiac death is ventricular fibrillation with underlying inflammation due to high cortisol concentration [9,76,77]. Long-term or extreme emotional stress may predispose patients to HPA axis dysfunction, with the resultant dysregulation of cortisol release causing ventricular fibrillation and sudden cardiac death [9].

#### 3.3.4. Stroke

Stroke is the dominant cause of long-term impairment and disability among adults in modern communities. A stroke normally happens when a blood clot prevents blood from flowing to a part of the brain. Consequently, brain cells that lack blood will start to die within minutes. Patients surviving a stroke are likely to experience cognitive, visual, and motor deficits, depending on the location and severity of the brain tissue damage [89]. Acute ischemic stroke acts as a stressor and thereby activates the HPA axis, leading to high levels of glucocorticoid [90]. Based on the systematic review by Barugh et al. (2014), cortisol levels are elevated in most patients for at least seven days after stroke and are within the normal range by three months. High levels of cortisol following stroke is correlated with increased dependency, morbidity, and mortality. However, at present, there is limited evidence to presume that these connections are independent of the severity of stroke [91].

Another study by Zi and Shuai (2013) suggested that cortisol can be used as an independent short-term prognostic marker of functional outcome and death in Chinese acute ischemic stroke patients [92]. This is consistent with the findings of previous studies that have reported that hypercortisolemia was associated with older age, greater severity of neurological impairment, larger ischemic lesions on computed tomography (CT) scan, and poorer prognosis (increased disability and mortality) in patients with stroke [93,94]. Furthermore, patients with stroke and elevated levels of cortisol are more vulnerable to adverse cardiac complications, which may result in higher mortality rates [95]. A poor prognosis after stroke is the development of infectious diseases associated with immune dysregulation due to neuroendocrine disorder after stroke [96]. Increased levels of cortisol make patients more susceptible to infections [97].

### 3.4. Drug Delivery

#### The Association between the Role of Cortisol and Potential Cardioprotective Agents

Cortisol levels were proven to influence the risks of cardiovascular events that are attributed to the actions on glucocorticoid (GR) and mineralocorticoid receptors (MRs) [98]. The cortisol concentration in our body depends on the activity of the hypothalamic–pituitary–adrenal (HPA) axis as well as the regulation of the 11β-hydroxysteroid dehydrogenase (11β-HSD) enzyme. 11β-HSD, which is an enzyme that plays an important role in the conversion of cortisone to active cortisol, exists in two isoforms: 11β-hydroxysteroid dehydrogenase type 1 (11β-HSD1) and 11β-hydroxysteroid dehydrogenase type 2 (11β-HSD2). 11β-HSD1 is more widely distributed than 11β-HSD2, especially in liver, brain, adipose, and other tissues. It was indicated that 11β-HSD1 mRNA is not affected by the circadian rhythm [99,100], and this is much more likely to reflect high 11β-HSD1 substrate levels at peak HPA axis activity. Indeed, 11β-HSD1 may contribute to normal circadian control of the HPA axis, suggesting a conserved role in HPA axis regulation [101]. To date, there is a paucity of literature on the evidence of cortisol metabolism by 11β-HSD1 in the heart and its activity in regulating heart function. To the best of our knowledge, this is the first review on the association between the role of cortisol on the cardiovascular system and potential therapeutic agents.

In recent years, 11β-HSD1 has been implicated to be a part of the pathogenesis of cardiovascular diseases such as myocardial infarction and atherosclerosis. This is in accordance with the evidence that has been reviewed above, in which excessive cortisol is associated with an increased risk of heart failure and other heart conditions [84,102]. On the other hand, previous studies have reported that mice with 11β-HSD1 deficiency showed an improved heart function following the recruitment of proreparative macrophages and increased proangiogenic signaling [103]. In a study done by Kipari et al. (2013), it was reported that 11β-HSD1 is associated with a reduction in plaque size in an atherosclerotic model of ApoE-deficient mice due to the inhibition of macrophages and T-cell infiltration of atherosclerotic lesions. The atheroprotective effects of 11β-HSD1 deficiency are mediated through both systemic and local vascular cell adhesion molecule (VCAM) mechanisms [104].

11β-HSD1 deficiency or inhibition has been of great interest in the cardiovascular field as it is reported to skew to a “cardioprotective” plasma lipid profile, shifts hepatic lipid metabolism from lipogenesis to fatty acid oxidation, and causes a preferential gain of peripheral adipose tissue at the expense of visceral tissue. With regard to this, 11β-HSD1 is a novel molecular target of interest for treating metabolic syndrome and type-2 diabetes mellitus [105]. Metabolic syndrome is known as an array of cardiovascular risk factors that include visceral obesity, insulin resistance, dyslipidemia, and hypertension. Elevated 11β-HSD1 activity can be seen in patients with these conditions [106]. Overexpression of 11β-HSD1 will, in turn, stimulate the expression of TNF-α and interleukin-6 (IL-6). This indicates that high levels of glucocorticoids promote an inflammatory response rather than an anti-inflammatory effect through cortisol [107,108]. These studies are consistent with the notion that 11β-HSD1 is the major determinant of cortisol excess and its inhibition will alleviate metabolic abnormalities.

The inflammatory response of cortisol is congruent to the fact that cardiovascular diseases are often related with proinflammatory cytokines such as IL-1, IL-6, and TNF-α. These cytokines are also key activators of 11β-HSD1 gene expression [109,110]. High concentrations of these cytokines will stimulate oxidative stress, downregulate endothelial nitric oxide synthase (eNOS) activity, and induce endothelial cell apoptosis, which will result in the deterioration in vascular function, as seen in conditions such as atherosclerosis, acute coronary syndrome, and heart failure. This is in accordance with the expanded use of statins (HMG-CoA reductase inhibitors) as anti-inflammatory agents in managing cardiovascular risks. Other than the available effective cardioprotective agents such as aspirin, anticoagulants, beta-blockers, angiotensin-converting enzyme inhibitors, calcium channel blockers, and many more, advanced research has begun to emphasize the relationship between cytokines and cardiovascular diseases in order to develop targeted anticytokine therapy.

A cohort study known as CANTOS (Canakinumab Anti-inflammatory Thrombosis Outcome Study) recently provided the first piece of evidence that targeting inflammation in humans with atherosclerosis could enhance clinical outcomes. It was proven that treatment with the anti–IL-1β antibody canakinumab significantly reduced recurrent cardiovascular events in individuals with stable coronary artery disease [111]. A few other studies also reported that canakinumab treatment was able to reduce cancer incidence and mortality, reduce gout attacks, and improve heart failure outcomes [112,113]. Other clinical studies have reported the protective effects of anti–TNF-α and anti–IL-6 receptor monoclonal antibodies on cardiovascular risk. However, further studies are warranted to elucidate the comprehensive mechanism behind the potential cardio-protective effects of these anti-cytokine agents. Overall, it is of great advantage to emphasize the association between cytokines and cardiovascular diseases as cytokines have a major impact on many inflammatory diseases. This association could potentially be used to develop a line of repurposable drugs.

It can be concluded that selective modulation of 11β-HSD1 activity in immune–inflammatory cells may provide a new therapeutic approach for cardiovascular diseases, consisting of both 11β-HSD1 inhibitors and anticytokine agents. The cardiotherapeutic potential of these agents may strengthen the effectiveness of currently available cardiovascular drugs. Table 1 summarizes the cardioprotective agents available worldwide.

## 4. Conclusions

In a nutshell, cortisol is a key player in the circadian system that significantly regulates cardiac function. Previous studies have emphasized that excessive cortisol is correlated with an increased risk of cardiovascular events, namely, acute coronary syndromes, arrhythmias, sudden cardiac death, and stroke. The present review highlights the importance of cortisol on circadian rhythm and its effect on the cardiovascular system, as well as potential therapeutic agents. Further studies are recommended to explore more cardioprotective agents that may be beneficial for treating cardiovascular complications, thereby reducing morbidity and mortality.

## Figures and Tables

**Figure 1 ijerph-18-00676-f001:**
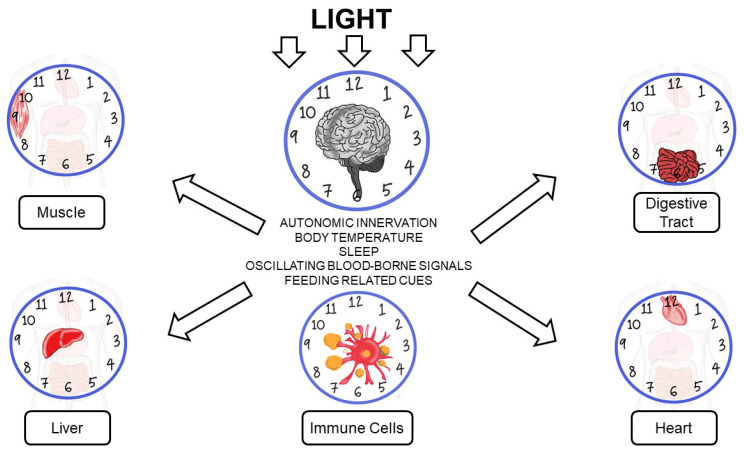
Biological clock in heart, digestive tract, liver, muscle, and immune cells.

**Figure 2 ijerph-18-00676-f002:**
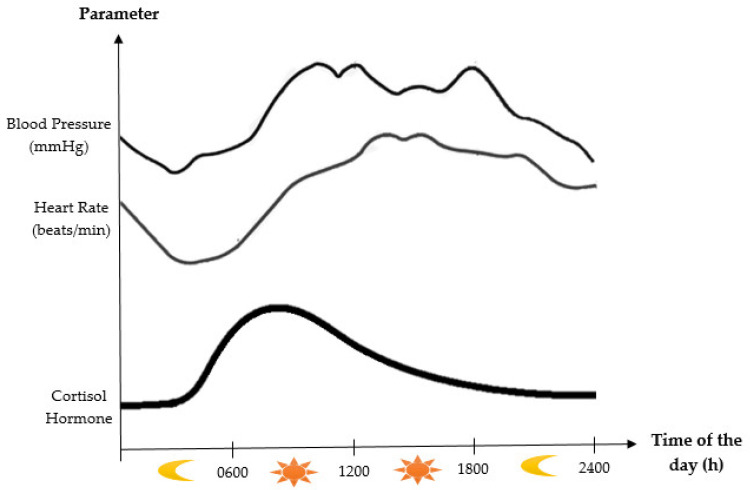
Diagram showing the circadian rhythms of blood pressure, heart rate, and cortisol hormone. Adapted from Selfridge et al. (2015) and Chan et al. (2010) [1,27].

**Table 1 ijerph-18-00676-t001:** The cardioprotective agents.

Study & Country	Drugs	Description of the Drugs	Example of Drugs
Antonopoulos et al. 2012, United Kingdom [114]	Statins or HMG-CoA reductase inhibitors	Potent lipid-lowering medication that inhibits biosynthesis of cholesterol. The anti-inflammatory properties of statins play important roles because of their protective effects on patients with coronary heart disease.	Atorvastatin, Simvastatin, Lovastatin
Fuster and Sweeny 2011, United States [115]	Aspirin	Efficacious anti-inflammatory, antiplatelet, and antithrombotic agent. Immediate treatment that is used in the management of patients with acute coronary syndromes and as a secondary preventive method.	Aspirin
De Caterina et al. 2012, Italy [116]	Anticoagulants	Therapy for prevention of stroke in atrial fibrillation, as well as secondary prevention after acute coronary syndromes.	Dabigatran Etexilate, Rivaroxaban, Apixaban
Rienstra et al. 2012, Netherlands [117]	Beta-blockers	Recommended drug in managing heart failure and atrial fibrillation, specifically for different indications.	Bisoprolol, Metoprolol, Carvedilol
Van Vark et al. 2012, Netherlands [118]	Angiotensin-converting enzyme inhibitors	The most commonly prescribed class of drugs for hypertension control, reducing cardiovascular morbidity and mortality.	Enalapril, Perindopril, Lisinopril
Elliott and Ram 2011, United States [119]	Calcium channel blockers	Drugs that result in vasodilation by productively lowering blood pressure across all patient populations.	Verapamil, Diltiazem, Nifedipine, Amlodipine
Ait-Oufella et al. 2019, France [111]	Anticytokine therapy	Reduces recurrent cardiovascular events in patients with stable coronary artery disease. CANTOS presented the first piece of evidence that targeting inflammation in humans with atherosclerosis could enhance clinical outcomes.	Anti-IL-1β antibody Canakinumab
Gray et al. 2017, United Kingdom [120]	11β-HSD1 inhibitors	As a therapeutic target for improving repair after myocardial infarction and preventing cardiac remodeling and heart failure from emerging.

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
