# Peer review of "Cortisol on Circadian Rhythm and Its Effect on Cardiovascular System"

_ijerph, 2021, doi:10.3390/ijerph18020676_

Round 1

Reviewer 1 Report

The authors reviewed about the relevance between circadian rhythm and effect of cortisol on cardiovascular system. Although the manuscript consisted of 3 sections and several sub sections, some of the section titles were inappropriate for review paper. For instance, the title of section 3; “Results and Discussion” is not suit for review paper. Moreover, there were several typos. The points are listed in <Comments to Authors>.

Reviewer 2 Report

The review summarizes nicely the role of cortisol, as well as that of relevant drugs targeting the 11β-HSD1 enzyme, on the cardiovascular system. It is an informative addition to current literature, relatively well written and adequately structured.

My main issue is with the abstract and most of the introduction. I find that these sections do not really aid the reader get to the main message of the review, and are rather confusing, and should thus be reorganized, with a focus on the cortisol-cardiovascular and cortisol-circadian interaction, instead of cortisol-sleep connection.

Additionally, if sleep behavior is considered so important for the review, as it occupies the first two paragraphs of the introduction, the relationship between sleep cycles and cardiovascular system, and/or disorders, should be explained more in depth. In the same direction, studies reporting the underlying mechanism of the cortisol-sleep or HPA-sleep interaction, should be mentioned in separate paragraphs. Otherwise, I would suggest removing the sleep aspect and focusing on the cortisol-circadian-cardiovascular system interplay.

Reviewer 3 Report

Author described the daily circadian rhythm of plasma cortisol level in 24 hour period and discussed the association of cortisol and cardiovascular diseases and the role of 11β-hydroxysteroid dehydrogenase (11β-HSD1) as a novel molecular target of interest for treating metabolic syndrome and type-2 diabetes mellitus. However, the present review did not highlight the role of cortisol which controls the circadian rhythm in sufficient detail as claimed in the title and abstract.

Major comment:

Cortisol certainly involved in stress-induced circadian disruption, including sleep disorder. In turn, circadian disruption by sleep disorder, shift work and jet-lag may also affect the circadian rhythm of cortisol involved in cardiovascular event. Therefore, it would be great addition to this review paper if author can discuss how cortisol control the circadian rhythm, what is the critical role of cortisol in the development of stress-induced sleep disorder (e.g., the association of cortisol and melatonin secretion at midnight), what is the current understanding about the potential mechanisms underlying the circadian pattern of cardiovascular event (e.g., heart stroke happens more often at early morning).

Minor comments:

All the association need to be clearly described if it is positive or negative association. E.g., line 191-193.

Round 2

Reviewer 3 Report

Thanks for the revision. This version was accepted for publication.